# Do community-level factors play a role in HIV self-testing uptake, linkage to services and HIV-related outcomes? A mixed methods study of community-led HIV self-testing in rural Zimbabwe

Mary K. Tumushime[1,2‡]*, Nancy Ruhode[1‡], Melissa Neuman[3], Constancia Watadzaushe[1], Miriam Mutseta[4], Miriam Taegtmeyer[5,6], Cheryl C. Johnson[7], Karin Hatzold[8], Elizabeth L. Corbett[9,10], Frances M. Cowan[1,11], Euphemia L. Sibanda[1,11]

1 Centre for Sexual Health and HIV/AIDS Research (CeSHHAR) Zimbabwe, Harare, Zimbabwe, 2 Faculty of Public Health and Policy, London School of Hygiene & Tropical Medicine (LSHTM), London, United Kingdom, 3 MRC International Statistics and Epidemiology Group, London School of Hygiene & Tropical Medicine (LSHTM), London, United Kingdom, 4 Population Services International (PSI) Zimbabwe, Harare, Zimbabwe, 5 Department of Clinical Sciences, Liverpool School of Tropical Medicine (LSTM), Liverpool, United Kingdom, 6 Tropical Infectious Disease Unit, Royal Liverpool University Hospital, Liverpool, United Kingdom, 7 Global HIV, Hepatitis and STI Programmes, World Health Organization, Geneva, Switzerland, 8 Population Services International (PSI), Washington, The District of Columbia, United States of America, 9 Malawi,-Liverpool Wellcome Trust Clinical Research Programme (MLW), Blantyre, Malawi, 10 Faculty of Infectious and Tropical Diseases, London School of Hygiene & Tropical Medicine (LSHTM), London, United Kingdom, 11 Department of International Public Health, Liverpool School of Tropical Medicine (LSTM), Liverpool, United Kingdom

‡ MKT and NR are co-first authors.
* Mary.Tumushime@lshtm.ac.uk

## Abstract

Community-led interventions, where communities plan and lead implementation, are increasingly being adopted within public health programmes. We explore factors associated with successful community-led distribution of HIV self-test (HIVST) kits to guide future service delivery. Twenty rural communities were supported to distribute HIVST kits for 1-month between January and September 2019. Social science researchers observed communities during planning and HIVST distribution, documenting findings in a standard observation template. Three months post-intervention, a population-based survey measured self-reported new HIV diagnosis, HIVST uptake, linkage to post-test services; and collected blood samples for viral load testing. The survey also included questions related to community cohesion; respondents' communities were grouped into low/medium/high based on community cohesion scores. We used mixed effect logistic regression to assess how outcomes differed based on community cohesion scores. In total, 27,812 kits were distributed by 348 distributors. Two HIVST distribution models were implemented: door-to-door only or at community venues/events. Of 5,683 participants surveyed, 1,831 (32.2%) received kits and 1,229 (67.1%) reported self-testing; overall HIVST uptake was 1,229/5,683 (21.6%). New HIV diagnosis increased with community cohesion, from

**Data availability statement:** The data and data dictionary have been shared as supplementary files.

**Funding:** This work was funded by Unitaid, sub agreement number 4214-CeSHHAR, awarded to ELC. The funder had no role in study design, data collection and analysis, decision to publish, or preparation of the manuscript.

**Competing interests:** The authors have declared that no competing interests exist.

32/1,770 (1.8%) in the low-cohesion group to 40/1,871 (2.1%) in the medium-cohesion group, adjusted odds ratio (aOR) 2.94 (1.41-6.12, p = 0.004) and 66/2,042 (3.2%) in the high-cohesion group, aOR 7.20 (2.31-22.50, p = 0.001). Other outcomes did not differ by extent of cohesion. Our findings demonstrate the more cohesive communities are, the more effective they may be at distributing HIVST kits and identifying people with undiagnosed HIV. Efforts to increase community cohesion should be considered as part of public health programmes and for planning and scaling-up HIVST implementation in communities.

## Introduction

HIV self-testing (HIVST) can increase testing coverage and frequency including among groups who may not otherwise test [1]. Since the release of the World Health Organization (WHO) guidelines in 2016, many countries have introduced HIVST to reach those in greatest need of HIV prevention and treatment. Since 2017, when HIVST was introduced in Zimbabwe, HIVST implementation has scaled up across facility and community settings, including through secondary distribution to partners and social contacts. However, as HIVST scale-up continues, there is a need to continue optimising distribution models to ensure they are reaching priority groups and achieving public health impact.

Historically, community-based HIVST programmes have demonstrated high impact on testing and linkage [2–4]. In Zimbabwe, community-based HIVST led by paid distributors achieved high testing uptake (50.3%), including among first-time testers (who comprised 36.3% of self-testers), young people under 25 years (46.2%) and men (46.5%) [2]. There was also a 27% increase in the uptake of antiretroviral therapy (ART) during distribution [2]. Despite its success, however, this HIVST distribution model was costly and resource-intensive [5]. Identifying sustainable service delivery models is critical when considering the growing challenges with testing efficiency and effectiveness in countries such as Zimbabwe, where there are fewer people with undiagnosed and untreated HIV, and progress toward global goals to end AIDS by 2030 among subpopulations is not uniform [6].

Community-led models, in which communities plan and lead the implementation or delivery of interventions [7], are increasingly being explored as a potentially more sustainable, lower cost and empowering approach to delivering public health interventions. Community-led interventions have been successfully adopted and effectively implemented across areas, such as in sanitation programmes [8], dengue prevention [9], and multi-disease campaigns including HIV, malaria, hypertension and diabetes screening [10]. Due to their effectiveness, global HIV targets now advocate for the involvement of communities in planning, delivery, and monitoring HIV interventions [11]. For instance, the Global AIDS strategy 2021–2026 advocates for community-led AIDS responses and calls for 30% of testing and treatment services, 80% of HIV prevention services, and 60% of societal enabler programmes to be led by local communities and/or community organisations [12]. However, HIVST approaches in Zimbabwe have yet to adopt community-led strategies fully.

Previous studies have highlighted how the success of community-led interventions is often attributed to strong local leadership and support, effective community mobilisation, community ownership, and encouraging people to have a whole-of-community rather than individual focus [8–10]. That each community can work together to customise their own interventions further strengthens this approach [9] and promotes community cohesion. Community cohesion, defined as the extent of connectedness and solidarity among groups in society [13], is associated with improved health outcomes [13,14] and can impact the success of community-led programmes. There is theoretical evidence to suggest that social identity and connectedness promote individual and group health behaviours, involvement in health-related community interventions and improved health outcomes [15–17]. The effect of community cohesion on HIV testing uptake in community-led interventions and subsequent linkage to prevention, treatment and care services has not been investigated.

In a trial conducted in Zimbabwe and reported separately [18], we determined the effect of community-led HIVST distribution on linkage to post-test services (confirmatory testing following reactive self-test results, voluntary medical male circumcision [VMMC] and pre-exposure prophylaxis [PrEP]) and self-reported recent/new HIV diagnosis. Each intervention community (cluster) in the trial was allowed to design and implement its own model of HIVST distribution. Here we explore the effects of the different community-led HIVST distribution models, levels of community involvement in planning distribution programmes and community cohesion on: (i) HIVST uptake, (ii) linkage to post-test services (confirmatory testing, VMMC and PrEP), and (iii) HIV-related outcomes (new HIV diagnosis and undetectable viral load). We hypothesised that communities in which distribution relied solely on distributors' efforts (i.e., only door-to-door), would perform poorer on outcomes i-iii above, compared to those in which community members actively sought and accessed HIVST kits. We also hypothesised that closely-knit or more cohesive communities would achieve better outcomes; see Fig 1 (conceptual framework).

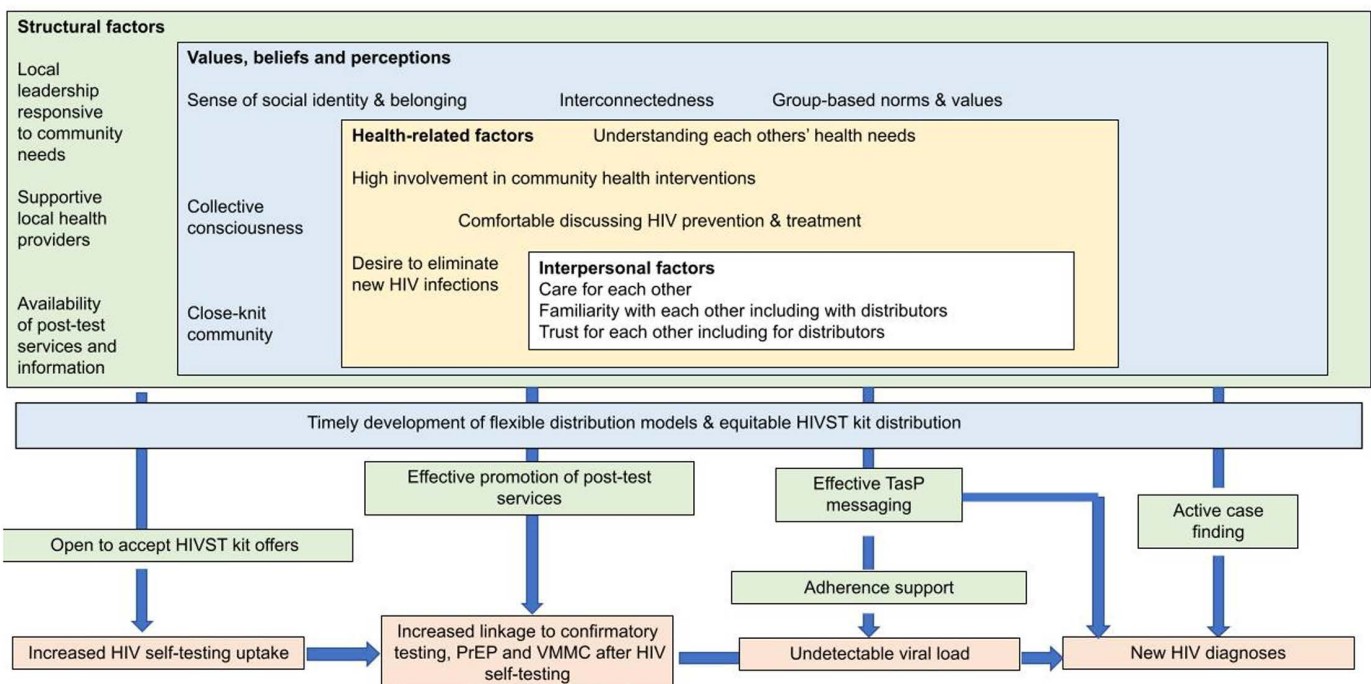

**Fig 1. Conceptual framework - Cohesive communities achieve better HIV-related outcomes.** PreP = pre-exposure prophylaxis; TasP = treatment as prevention; VMMC = voluntary medical male circumcision.

## Methods

### Setting

This study was conducted as part of the HIV Self-Testing Africa (STAR) Initiative (https://unitaid.org/project/self-testing-af-rica-star/#en), the largest evaluation of HIVST in Africa to date, that sought to catalyse the market for HIVST and drive global scale-up. The work presented here was nested within a cluster randomised trial in rural Zimbabwe which compared HIV testing and linkage outcomes between community-led (intervention) and community-based HIVST distribution led by paid distributors (comparison) [18]. Clusters/communities were defined as groups of adjacent villages headed by a local leader known as a headman (headman unit) and separated by at least 20 km. These were rural farming and mining communities served by a district hospital and/or rural health centre. Forty headman units in 6 districts were randomised 1:1 to the study arms. This paper focuses on the intervention arm.

**Implementation of the community-led intervention.** Community-led HIVST distribution was implemented in 20 headman units. In each headman unit, Population Services International (PSI) Zimbabwe conducted community engagement activities over a 2–3 week period to introduce the concept of community-led HIVST and to inform community members that ensuring people living with HIV were on treatment could ultimately reduce the number of new infections in their villages. Community engagement included (promotion of "U=U" messages (Undetectable = Untransmittable), where people learned those with an HIV viral load below the limit of detection will not transmit HIV to their partners) [19]. We packaged HIVST with U = U messaging as part of the intervention. We expected that knowledge of treatment as prevention (TasP) and that early ART treatment can reduce HIV morbidity and mortality, would prompt community members to seek HIV testing and treatment. Headman units were then invited to participate in the study and asked to design their own models of HIVST distribution. Decisions about distributor selection, where and how to access kits and/or distribute kits and provision of distributor incentives were driven by community members in the headman units. HIVST distribution models were therefore allowed to vary across the 20 communities. HIVST distributors were trained to promote and support HIVST, including how to deliver messages to encourage uptake of confirmatory testing and ART for those with a reactive (positive) self-test result and VMMC and PrEP for those with a non-reactive (negative) self-test result. Local health facilities managed bulk kit storage and supply and provided post-test services. While confirmatory testing for HIV was available from all health facilities, not all facilities provided VMMC and PrEP; in such cases, referrals were made. Headman units were advised to implement HIVST according to national guidance e.g., following minimum age requirements and ensuring self-testing was voluntary and non-coercive. In all headman units, distribution proceeded for 4 weeks. Headman units were given posters and flyers designed by PSI Zimbabwe to advertise HIVST availability and U = U. All communities then decided how best to use these materials.

Based on observations (described below), headman units implemented the following HIVST distribution models: (i) door-to-door only or (ii) a combination of different delivery approaches including door-to-door and collection of kits directly from distributors at their homes or at various locations in the headman unit (combined HIVST distribution model). The latter model was more participatory and headman units would refine their models iteratively. Changes included community members (including distributors) forming committees to provide ongoing planning and logistical support for distribution; distributors forming pairs or groups to support each other; distributors working in villages they had not initially been assigned to, to increase coverage; and distributors taking advantage of community gatherings (e.g., meetings and sporting events) and workplaces (e.g., markets, shopping centres and mines) to distribute self-test kits.

### Data collection

**Participant observations.** Participant observations were conducted by trained qualitative researchers at and between community sensitisation and planning meetings, during distributor training and during kit distribution in each headman unit to explore their progress and as part of process evaluation. Observation findings were documented using a template

 

which captured levels of attendance and diversity of attendees (men, young people, leaders) at sensitisation meetings, levels of participation at planning meetings, how decisions were made (whether through consensus or by coercion from community leaders), degree to which headman units appeared cohesive and aware of community-led HIVST distribution and/or HIVST (ascertained through informal discussions with community members, leaders, and healthcare workers), and presence in the community of promotional material (posters and flyers) about the intervention and HIVST.

**Population-based survey.** We conducted a representative population-based survey in randomly-selected households from three months after the end of HIVST distribution in each headman unit. First, we randomly selected three enumeration areas in each headman unit, followed by random selection of one in two households. All individuals aged ≥16 years in selected households were invited to participate. The questionnaire was self-administered in the preferred language (English or two major local languages) using audio-computer assisted survey instrument (ACASI). Participants were asked about household and individual demographic characteristics, HIV testing history, experiences with HIV self-testing and linkage to post-test services. Participants were also asked to respond to a six-item measure of community cohesion validated by Lippman et al. in high HIV prevalence settings in South Africa: (i) people in this community are willing to help their neighbours, (ii) this is a close knit community, (iii) people in this community can be trusted, (iv) people in this community get along well with each other, (v) people in this community share the same values and (vi) people in this community look out for each other [20]. Cohesion in this context refers to individuals' perceptions of the extent to which their community is closely knit, shares the same values, trusts each other and members are willing to help each other. All items had response options of strongly agree, somewhat agree, neither agree nor disagree, somewhat disagree, and strongly disagree. Item response modelling (IRM) was used to assess and summarise the cohesion scale using a validated, one-parameter multinomial model [20]. To verify self-reports of HIV status and measure viral load, dried blood spot (DBS) samples were taken to test for HIV and viral load.

## Outcomes

Outcomes were based on self-reports among survey respondents and viral load results. The following outcomes were compared between (i) levels of community cohesion (ii) the two distribution models headman units employed, and (iii) levels of community involvement:

1. Proportion of participants self-reporting uptake of HIVST. The numerator was the number of surveyed participants self-reporting they used an HIV self-test kit to test themselves, the denominator was the total number of surveyed participants.

2. Proportion of participants self-reporting linkage to post-test services following HIVST (combined and individual measures):

    i) Confirmatory testing: the numerator was the number of surveyed participants self-reporting uptake of confirmatory testing following a reactive (HIV positive) self-test result, the denominator was the total number of surveyed participants who self-reported a reactive self-test result

    ii) VMMC: the numerator was the number of male participants self-reporting uptake of VMMC following a non-reactive (HIV negative) self-test result, the denominator was the number of male participants who self-reported a non- reactive self-test result

    iii) PrEP: the numerator was the number of participants self-reporting uptake of PrEP following a non-reactive (HIV negative) self-test result, the denominator was the number of surveyed participants who self-reported a non-reactive self- test result

3. New HIV diagnosis following HIV self-testing. New HIV diagnosis was defined as a self-reported, new provider-confirmed positive test since the start of HIVST distribution. The numerator was the number of surveyed participants reporting a new HIV diagnosis, the denominator was the total number of surveyed participants.

4. Undetectable viral load among people living with HIV (<1,000 copies/ml). The numerator was the number of surveyed participants with undetectable viral load, the denominator was the total number of HIV positive participants, as determined through laboratory testing of DBS samples.

## Data analysis

**Participant observations.** We described levels of community involvement using six attributes shown in Table 1, each with three categories agreed by the research team. Information pertaining to these attributes was detailed in the template described above for each headman unit. Two researchers reviewed observation reports and independently scored headman units on each attribute, with scores ranging from 1-3, with the lowest score indicating the lowest level of community involvement. Discrepancies were discussed and resolved with consensus. Scores were totalled for each headman unit. Headman units were then categorised by terciles (low, medium, and high) indicating their level of community involvement [21]. Construction of the scale which guided categorisation was done at the beginning, and cut-off points were based on ranks, as the measure was not normally distributed. For each headman unit, distribution models were coded (1) door-to-door distribution only or (2) combined HIVST distribution model. Community involvement scores and coded distribution models were merged with survey responses.

**Population-based survey.** Participants responded to a six-item measure of community cohesion (described above). Individual cohesion scores were calculated using the average of item responses ranging from 1 (strongly agree) to 5 (strongly disagree). Community cohesion was summarised as the median score of individuals within each headman unit, and headman units were then categorised by terciles (low, medium, and high cohesion) [21]. Construction of the scale is as above.

We used mixed effect logistic regression to assess how the outcomes above differed by distribution model, levels of community involvement and community cohesion. Analysis used Stata v14. Before analysis, we compared similarity by distribution model, levels of community involvement and cohesion group for pre-specified variables to identify substantial differences needing adjustment. All outcomes were analysed using mixed effect logistic regression. All models are

**Table 1. Attributes from participant observations used to determine levels of community involvement in planning distribution programmes.**

| Attribute | Score | | |
|---|---|---|---|
| | 1 | 2 | 3 |
| 1. Attendance at first community sensitization meeting | Low | Medium | High |
| 2. Diversity of attendees at first community sensitisation meeting | | | |
| Proportion of men | Low | Medium | High |
| Proportion of young people | Low | Medium | High |
| Proportion of leaders | Least favourable | Medium | Most favourable |
| 3. Participation in planning processes | Decisions dominated by leaders/few key individuals | Moderate participation | Majority participating |
| 4. Ability of headman unit to finalise distribution model in a timely fashion | More than 3 weeks' time | Within 2–3 weeks' time | Within 1 week |
| 5. Awareness of HIVST and CLD | Poor knowledge of HIVST or CLD | Some awareness of HIVST and/or CLD | Widespread awareness of both interventions |
| 6. Support given by headman unit to distributors | No evidence of support given | Some evidence of support | Clear evidence of support |

CLD=community led HIVST distribution; HIVST=HIV self-testing.

adjusted for district, age, gender, and educational attainment. All models adjusted for the study community using a random effect. Fisher's exact test was used to determine if there were significant associations between the community level measures: (i) distribution model and levels of community involvement, (ii) distribution model and community cohesion, and (iii) levels of community involvement and community cohesion.

## Ethical considerations

Ethical approval to conduct this study was obtained from the Medical Research Council of Zimbabwe (ref. MRCZ/A/2323), London School of Hygiene and Tropical Medicine Research Ethics Committee (ref. 15801–1) and World Health Organization Research Ethics Review Committee (ref. ERC.0003065). The trial was registered with Pan African Clinical Trial Registry, ref PACTR201811849455568. Written informed consent was obtained from participants prior to the survey and collection of DBS.

The funder, Unitaid, had no role in study design, data collection and analysis, decision to publish, or preparation of the manuscript.

## Results

The researchers included the data (S1 Data) and data dictionary (S2 Data) as supporting information.

### Implementation of community-led HIVST distribution models and survey response rate

Implementation of the community-led HIVST distribution models (inclusive of community engagement, HIVST distribution and the survey) was conducted between October 2018 and December 2019, in 20 headman units randomised to the community-led HIVST distribution arm. Five headman units (5/20, 25.0%) in 3 study districts only offered HIVST door-to-door (Table 2). In the other 15 headman units (15/20, 75.0%) across 6 study districts, a combination of different delivery approaches was used including door-to-door distribution combined with collection of kits directly from distributors at their homes or at various locations in the headman unit (combined HIVST distribution model). Overall, 348 distributors were trained and distributed 27,812 kits, with a range of 28–159 kits distributed per distributor.

Participant observations indicated joint messaging on HIVST and U=U was widely disseminated and well received during community engagement. Headman units actively participated in the design and implementation of their distribution models and were well-supported by local leaders before and during distribution.

From 3,000 households in headman units implementing community-led HIVST distribution, 5,683/6,748 eligible participants were surveyed, with a response rate of 84.2%. Among surveyed participants, 1,831 (32.2%) received a self-test kit of whom 1,229/1,831 (67.1%) reported self-testing, giving an overall HIVST uptake of 21.6% (1,229/5,683). Uptake did not differ by distribution model (door-to-door: 358/1,542 (23.2%) vs combined: 871/4,141 (21.0%); (adjusted odds ratio [aOR]: 0.98 (95% confidence interval [CI]: 0.64-1.51, p=0.92).

### Participant and community characteristics

Tables 2–4 show cluster- and individual-level characteristics of participants in the programme and population-based survey by distribution model, levels of community involvement, and community cohesion respectively. Participant characteristics were largely comparable by distribution model, levels of community involvement and cohesion group. Of 5,683 participants surveyed, over half (54.6%) were female. The majority were aged 20–29 years (26.1%), married or cohabitating (60.0%) and had completed secondary education or higher (38.8%). Headman units ranged from 4-36 villages in size (12.4 villages per cluster).

For community involvement in planning HIVST distribution programmes, 2 headman units were classified as low involvement, 11 were classified as medium involvement and 7 high involvement (Table 3). Headman units in the low involvement group were in 2 study districts, while those in medium and high involvement groups were spread across 5 study districts each.

**Table 2. Cluster- and individual-level characteristics of participants in programme and population-based survey by HIVST distribution model.**

| Characteristics | Door-to-door only | | Door-to-door plus household/community collection | | Total | |
|---|---|---|---|---|---|---|
| | n/n | % | n/n | % | n/N | % |
| **Cluster-level** | | | | | | |
| **Total** | 5 | 100.0% | 15 | 100.0% | 20 | 100.0% |
| District | | | | | | |
| Mutoko | 0/5 | 0.0% | 4/15 | 26.7% | 4/20 | 20.0% |
| Muzarabani | 0/5 | 0.0% | 2/15 | 13.3% | 2/20 | 10.0% |
| Shamva | 1/5 | 20.0% | 2/15 | 13.3% | 3/20 | 15.0% |
| Shurugwi | 2/5 | 40.0% | 2/15 | 13.3% | 4/20 | 20.0% |
| Umguza | 0/5 | 0.0% | 2/15 | 13.3% | 2/20 | 10.0% |
| Zvimba | 2/5 | 40.0% | 3/15 | 20.0% | 5/20 | 25.0% |
| Villages per cluster (mean/SD) | 9.8 (5.4) | | 13.3 (10.5) | | 12.4 (9.5) | |
| **Individual-level** | | | | | | |
| **Total** | 1542 | 100.0% | 4141 | 100.0% | 5683 | 100.0% |
| Female | 885/1542 | 57.4% | 2218/4141 | 53.6% | 3103/5683 | 54.6% |
| Age in groups | | | | | | |
| 16-19 years | 252/1542 | 16.3% | 673/4141 | 16.3% | 925/5683 | 16.3% |
| 20-29 years | 430/1542 | 27.9% | 1054/4141 | 25.5% | 1484/5683 | 26.1% |
| 30-39 years | 355/1542 | 23.0% | 913/4141 | 22.0% | 1268/5683 | 22.3% |
| 40-49 years | 281/1542 | 18.2% | 635/4141 | 15.3% | 916/5683 | 16.1% |
| 50-59 years | 118/1542 | 7.7% | 352/4141 | 8.5% | 470/5683 | 8.3% |
| 60 + years | 105/1542 | 6.8% | 510/4141 | 12.3% | 615/5683 | 10.8% |
| Marital status* | | | | | | |
| Married or living as married | 1002/1528 | 65.6% | 2362/4073 | 58.0% | 3364/5601 | 60.0% |
| Never married | 334/1528 | 21.9% | 1032/4073 | 25.3% | 1366/5601 | 24.4% |
| Widowed/separated/divorced | 192/1528 | 12.6% | 679/4073 | 16.7% | 871/5601 | 15.6% |
| Highest level of education | | | | | | |
| Primary complete or less | 677/1542 | 43.9% | 1361/4141 | 32.9% | 2038/5683 | 35.9% |
| Some secondary education | 383/1542 | 24.8% | 1058/4141 | 25.5% | 1441/5683 | 25.4% |
| Secondary education complete or higher | 482/1542 | 31.3% | 1722/4141 | 41.6% | 2204/5683 | 38.8% |
| Religion | | | | | | |
| Apostolic | 644/1542 | 41.8% | 1411/4141 | 34.1% | 2055/5683 | 36.2% |
| Other | 898/1542 | 58.2% | 2730/4141 | 65.9% | 3628/5683 | 63.8% |
| Receives regular salary† | 364/1529 | 23.8% | 899/4089 | 22.0% | 1263/5618 | 22.5% |

*14 missing marital status in communities with door-to-door only distribution and 68 in communities with door-to-door plus household/community collection.

†13 missing salary data in communities with door-to-door only distribution and 52 in communities with door-to-door plus household/community collection.

PLHIV = people living with HIV; PrEP = pre-exposure prophylaxis; SD = standard deviation; VMMC = voluntary medical male circumcision.

For community cohesion, 6 headman units were in the low cohesion group and 7 headman units each in medium and high cohesion, respectively (Table 4). Headman units in low and high cohesion groups were in 3 study districts, while those in the medium cohesion group were in 5 study districts.

**Table 3. Cluster- and individual-level characteristics of participants in programme and population-based survey by level of community involvement in planning distribution programmes.**

| Characteristics | Low involvement | | Middle involvement | | High involvement | | Total | |
|---|---|---|---|---|---|---|---|---|
| | n/n | % | n/n | % | n/n | % | n/n | % |
| **Cluster-level** | | | | | | | | |
| **Total** | 2 | 100.0% | 11 | 100.0% | 7 | 100.0% | 20 | 100.0% |
| District | | | | | | | | |
| Mutoko | 1/2 | 50.0% | 1/11 | 9.1% | 2/7 | 28.6% | 4/20 | 20.0% |
| Muzarabani | 0/2 | 0.0% | 1/11 | 9.1% | 1/7 | 14.3% | 2/20 | 10.0% |
| Shamva | 0/2 | 0.0% | 1/11 | 9.1% | 2/7 | 28.6% | 3/20 | 15.0% |
| Shurugwi | 0/2 | 0.0% | 3/11 | 27.3% | 1/7 | 14.3% | 4/20 | 20.0% |
| Umguza | 1/2 | 50.0% | 0/11 | 0.0% | 1/7 | 14.3% | 2/20 | 10.0% |
| Zvimba | 0/2 | 0.0% | 5/11 | 45.5% | 0/7 | 0.0% | 5/20 | 25.0% |
| Villages per cluster (mean/SD) | 15.0 (11.3) | | 11.7 (9.0) | | 12.7 (11.1) | | 12.4 (9.5) | |
| **Individual-level** | | | | | | | | |
| **Total** | 588 | 100.0% | 3080 | 100.0% | 2015 | 100.0% | 5683 | 100.0% |
| Female | 303/588 | 51.5% | 1709/3080 | 55.5% | 1091/2015 | 54.1% | 3103/5683 | 54.6% |
| Age in groups* | | | | | | | | |
| 16-19 years | 95/588 | 16.2% | 468/3077 | 15.2% | 362/2013 | 18.0% | 925/5678 | 16.3% |
| 20-29 years | 167/588 | 28.4% | 798/3077 | 25.9% | 519/2013 | 25.8% | 1484/5678 | 26.1% |
| 30-39 years | 154/588 | 26.2% | 662/3077 | 21.5% | 452/2013 | 22.5% | 1268/5678 | 22.3% |
| 40-49 years | 76/588 | 12.9% | 521/3077 | 16.9% | 319/2013 | 15.8% | 916/5678 | 16.1% |
| 50-59 years | 48/588 | 8.2% | 271/3077 | 8.8% | 151/2013 | 7.5% | 470/5678 | 8.3% |
| 60+years | 48/588 | 8.2% | 357/3077 | 11.6% | 210/2013 | 10.4% | 615/5678 | 10.8% |
| Marital status† | | | | | | | | |
| Married or living as married | 314/575 | 54.6% | 1829/3042 | 60.1% | 1221/1984 | 61.5% | 3364/5601 | 60.1% |
| Never married | 175/575 | 30.4% | 697/3042 | 22.9% | 494/1984 | 24.9% | 1366/5601 | 24.4% |
| Widowed/separated/divorced | 86/575 | 15.0% | 516/3042 | 17.0% | 269/1984 | 13.6% | 871/5601 | 15.6% |
| Highest level of education | | | | | | | | |
| Primary complete or less | 115/588 | 19.6% | 1137/3080 | 36.9% | 786/2015 | 39.0% | 2038/5683 | 35.9% |
| Some secondary education | 128/588 | 21.8% | 763/3080 | 24.8% | 550/2015 | 27.3% | 1441/5683 | 25.4% |
| Secondary education complete or higher | 345/588 | 58.7% | 1180/3080 | 38.3% | 679/2015 | 33.7% | 2204/5686 | 38.8% |
| Religion | | | | | | | | |
| Apostolic | 147/588 | 25.0% | 1096/3080 | 35.6% | 812/2015 | 40.3% | 2055/5683 | 36.2% |
| Other | 441/588 | 75.0% | 1984/3080 | 64.4% | 1203/2015 | 59.7% | 3628/5683 | 63.8% |
| Receives regular salary# | 213/582 | 36.6% | 654/3046 | 21.5% | 396/1990 | 19.9% | 1263/5618 | 22.5% |

*3 missing age in medium involvement group and 2 in high involvement group.

†13 missing marital status in low involvement group, 38 medium involvement group and 31 in high involvement group.

#6 missing salary data in low involvement group, 34 medium involvement group and 25 in high involvement group.

PLHIV=people living with HIV; PrEP=pre-exposure prophylaxis; SD=standard deviation; VMMC=voluntary medical male circumcision.

## Effect of community cohesion on outcomes

Self-report of a new HIV diagnosis increased with community cohesion, from 32/1,770 (1.8%) in the lowest cohesion group to 40/1,871 (2.1%) in the medium group, aOR 2.94 (1.41-6.12, p=0.004) and 66/2,042 (3.2%) in the highest cohesion group, aOR 7.20 (2.31-22.50, p=0.001) (Table 5).

**Table 4. Cluster- and individual-level characteristics of participants in programme and population-based survey by self-reported community cohesion.**

| Characteristics | Low cohesion | | Middle cohesion | | High cohesion | | Total | |
|---|---|---|---|---|---|---|---|---|
| | n/n | % | n/n | % | n/n | % | n/n | % |
| **Cluster-level** | | | | | | | | |
| **Total** | 6 | 100.0% | 7 | 100.0% | 7 | 100.0% | 20 | 100.0% |
| District | | | | | | | | |
| Mutoko | 1/6 | 16.7% | 1/7 | 14.3% | 0/7 | 0.0% | 2/20 | 10.0% |
| Muzarabani | 0/6 | 0.0% | 2/7 | 28.6% | 1/7 | 14.3% | 3/20 | 15.0% |
| Shamva | 0/6 | 0.0% | 2/7 | 28.6% | 2/7 | 28.6% | 4/20 | 20.0% |
| Shurugwi | 4/6 | 66.7% | 1/7 | 14.3% | 0/7 | 0.0% | 5/20 | 25.0% |
| Umguza | 1/6 | 16.7% | 1/7 | 14.3% | 0/7 | 0.0% | 2/20 | 10.0% |
| Zvimba | 0/6 | 0.0% | 0/7 | 0.0% | 4/7 | 57.1% | 4/20 | 20.0% |
| Villages per cluster (mean/SD) | 8 (2.8) | | 10.3 (11.5) | | 18.3 (8.9) | | 12.4 (9.5) | |
| **Individual-level** | | | | | | | | |
| **Total** | 1770 | 100.0% | 1871 | 100.0% | 2042 | 100.0% | 5683 | 100.0% |
| Female | 992/1770 | 56.0% | 947/1871 | 50.6% | 1164/2042 | 57.0% | 3103/5683 | 54.6% |
| Age in groups | | | | | | | | |
| 16-19 years | 263/1770 | 14.9% | 326/1871 | 17.4% | 336/2042 | 16.5% | 925/5683 | 16.3% |
| 20-29 years | 475/1770 | 26.8% | 477/1871 | 25.5% | 532/2042 | 26.1% | 1484/5683 | 26.1% |
| 30-39 years | 439/1770 | 24.8% | 440/1871 | 23.5% | 389/2042 | 19.0% | 1268/5683 | 22.3% |
| 40-49 years | 293/1770 | 16.6% | 293/1871 | 15.7% | 330/2042 | 16.2% | 916/5683 | 16.1% |
| 50-59 years | 142/1770 | 8.0% | 144/1871 | 7.7% | 184/2042 | 9.2% | 470/5683 | 8.3% |
| 60+years | 158/1770 | 8.9% | 189/1871 | 10.1% | 268/2042 | 13.1% | 615/5683 | 10.8% |
| Marital status* | | | | | | | | |
| Married or living as married | 1093/1748 | 62.5% | 1140/1840 | 62.0% | 1131/2013 | 56.2% | 3364/5601 | 60.1% |
| Never married | 409/1748 | 23.4% | 447/1840 | 24.3% | 510/2013 | 25.3% | 1366/5601 | 24.4% |
| Widowed/separated/divorced | 246/1748 | 14.1% | 253/1840 | 13.8% | 372/2013 | 18.5% | 871/5601 | 15.6% |
| Highest level of education | | | | | | | | |
| Primary complete or less | 612/1770 | 34.6% | 743/1871 | 39.7% | 683/2042 | 33.4% | 2038/5683 | 35.9% |
| Some secondary education | 405/1770 | 22.9% | 515/1871 | 27.5% | 521/2042 | 25.5% | 1441/5683 | 25.4% |
| Secondary education complete or higher | 753/1770 | 42.5% | 613/1871 | 32.8% | 838/2042 | 41.0% | 2204/5683 | 38.8% |
| Religion | | | | | | | | |
| Apostolic | 646/1770 | 36.5% | 746/1871 | 39.9% | 663/2042 | 32.5% | 2055/5683 | 36.2% |
| Other | 1124/1770 | 63.5% | 1125/1871 | 60.1% | 1379/2042 | 67.5% | 3628/5683 | 63.8% |
| Receives regular salary† | 409/1754 | 23.3% | 393/1846 | 21.3% | 461/2018 | 22.8% | 1263/5618 | 22.5% |

*22 missing marital status in low cohesion group, 31 medium cohesion group and 29 in high cohesion group.

†16 missing salary data in low cohesion group, 25 medium cohesion group and 24 in high cohesion group.

PLHIV = people living with HIV; PrEP = pre-exposure prophylaxis; SD = standard deviation; VMMC = voluntary medical male circumcision.

Other study outcomes did not differ by level of cohesion. Cohesion had no overall effect on HIVST uptake across groups (p = 0.42); 451/1,770 (25.5%) participants in the low-cohesion group and 75/1,871 (20.0%) in the medium-cohesion group (aOR 0.60 (0.36-0.99), p = 0.05) reported uptake. In the high-cohesion group, 403/2,042 (19.7%) participants (aOR 0.63 (0.29-1.35), p = 0.23) reported HIVST uptake. Trend analysis using a linear parameterisation of the cohesion group variable showed there was no trend in cohesion and HIVST uptake; (aOR for 1-unit increase in cohesion score: 0.76 (95%CI: 0.51, 1.14), p = 0.182). Similarly, there were no differences in linkage to post-test services across groups, with

**Table 5. Comparison of outcomes by levels of community cohesion.**

| Outcome | | | Odds ratio (95% CI) | p-value | Adjusted odds ratio (95% CI) | p-value |
|---|---|---|---|---|---|---|
| | n/n | % | | | | |
| **Uptake outcome:** Self-reported self-testing uptake (N = 5683) | | | | | | |
| Low cohesion | 451/1770 | 25.5% | | | | |
| Medium cohesion | 375/1871 | 20.0% | 0.66 (0.40, 1.06) | 0.09 | 0.60 (0.36, 0.99) | 0.05 |
| High cohesion | 403/2042 | 19.7% | 0.71 (0.44, 1.15) | 0.17 | 0.63 (0.29, 1.35) | 0.23 |
| **Combined linkage outcome**: Self-reported linkage to confirmatory testing, VMMC and PrEP (N = 1229) | | | | | | |
| Low cohesion | 104/451 | 23.1% | | | | |
| Medium cohesion | 94/375 | 25.1% | 0.90 (0.53, 1.52) | 0.70 | 0.70 (0.40, 1.22) | 0.21 |
| High cohesion | 120/403 | 29.8% | 1.36 (0.83, 2.23) | 0.22 | 0.77 (0.37, 1.62) | 0.50 |
| **Linkage outcome 1:** Self-reported linkage to confirmatory testing (N = 1229) | | | | | | |
| Low cohesion | 9/451 | 2.0% | | | | |
| Medium cohesion | 7/375 | 1.9% | 0.93 (0.34, 2.53) | 0.89 | – | – |
| High cohesion | 15/403 | 3.7% | 1.90 (0.82, 4.39) | 0.13 | – | – |
| **Linkage outcome 2:** Self-reported linkage to VMMC (N = 1229) | | | | | | |
| Low cohesion | 39/451 | 8.6% | | | | |
| Medium cohesion | 47/375 | 12.5% | 0.86 (0.38, 1.94) | 0.72 | – | – |
| High cohesion | 36/403 | 8.9% | 0.98 (0.45, 2.14) | 0.96 | – | – |
| **Linkage outcome 3:** Self-reported linkage to PrEP (N = 1229) | | | | | | |
| Low cohesion | 67/451 | 14.9% | | | | |
| Medium cohesion | 61/375 | 16.3% | 1.00 (0.61, 1.64) | 1.00 | – | – |
| High cohesion | 86/403 | 21.3% | 1.51 (0.96, 2.38) | 0.08 | – | – |
| **HIV outcome 1:** Proportion of individuals reporting a new HIV diagnosis (N = 5683) | | | | | | |
| Low cohesion | 32/1770 | 1.8% | | | | |
| Medium cohesion | 40/1871 | 2.1% | 1.14 (0.54, 2.42) | 0.73 | 2.94 (1.41,6.12) | 0.004 |
| High cohesion | 66/2042 | 3.2% | 1.92 (0.93, 3.95) | 0.08 | 7.20 (2.31, 22.50) | 0.001 |
| **HIV outcome 2:** Undetectable viral load among PLHIV (N = 830) | | | | | | |
| Low cohesion | 150/227 | 66.1% | | | | |
| Medium cohesion | 135/234 | 57.7% | 0.71 (0.42, 1.23) | 0.22 | 0.93 (0.52, 1.66) | 0.81 |
| High cohesion | 224/369 | 60.7% | 0.82 (0.49, 1.37) | 0.44 | 0.79 (0.33, 1.89) | 0.60 |

PLHIV = people living with HIV; PrEP = pre-exposure prophylaxis; VMMC = voluntary medical male circumcision.

Adjusted odds ratios are not presented for the specific linkage outcomes (Linkage outcomes 1–3) due to the small number of cases. Adjusted models are adjusted for district and respondent age in 10-year groups, sex, and educational attainment (no or primary education, some secondary education, completed secondary education.). All models adjusted for study community using a random effect.

104/451 (23.1%) participants linking in the low-cohesion group, 94/375 (25.1%) in the medium-cohesion group, (aOR 0.70 (0.40-1.22), p = 0.21) and 120/403 (29.8%) in the high-cohesion group (aOR 0.77 (0.37-1.62), p = 0.50). Finally, undetectable viral load which was 150/227 (66.1%) participants in the low group, 135/234 (57.7%) in the medium group, (aOR 0.93 (0.52-1.66), p = 0.81) and 224/369 (60.7%) in the high group (aOR 0.79 (0.33-1.89), p = 0.60) and did not differ.

## Effect of HIVST distribution model on outcomes

Study outcomes did not differ by distribution model (Table 6).

New HIV diagnosis was reported by 157/4,141 (3.8%) and 54/1,542 (3.5%) participants where combined or door-to-door distribution models were implemented, respectively (aOR 1.42 (95% CI: 0.79-2.54), p = 0.24).

HIVST uptake in headman units implementing combined HIVST distribution models was reported by 871/4,141 (21.0%) participants and by 358/1,542 (23.2%) in headman units implementing door-to-door HIVST distribution only (aOR 0.98

**Table 6. Comparison of outcomes by distribution model.**

| Outcome | n/n | % | Odds ratio (95% CI) | p-value | Adjusted odds ratio (95% CI) | p-value |
|---|---|---|---|---|---|---|
| **Uptake outcome:** Self-reported self-testing uptake (N=5683) | | | | | | |
| Door-to-door only | 358/1542 | 23.2% | | | | |
| Door-to-door plus household/community collection | 871/4141 | 21.0% | 0.87 (0.54, 1.40) | 0.57 | 0.98 (0.64, 1.51) | 0.92 |
| **Combined linkage outcome:** Self-reported linkage to confirmatory testing, VMMC and PrEP (N=1229) | | | | | | |
| Door-to-door only | 101/358 | 28.2% | | | | |
| Door-to-door plus household/community collection | 217/871 | 24.9% | 0.80 (0.50, 1.28) | 0.35 | 0.92 (0.66, 1.27) | 0.60 |
| **Linkage outcome 1:** Self-reported linkage to confirmatory testing (N=1229) | | | | | | |
| Door-to-door only | 11/358 | 3.1% | | | | |
| Door-to-door plus household/community collection | 20/871 | 2.3% | 0.74 (0.35, 1.56) | 0.43 | – | – |
| **Linkage outcome 2:** Self-reported linkage to VMMC (N=1229) | | | | | | |
| Door-to-door only | 31/358 | 8.7% | | | | |
| Door-to-door plus household/community collection | 91/871 | 10.4% | 0.94 (0.46, 1.94) | 0.88 | – | – |
| **Linkage outcome 3:** Self-reported linkage to PrEP (N=1229) | | | | | | |
| Door-to-door only | 75/358 | 20.9% | | | | |
| Door-to-door plus household/community collection | 139/871 | 16.0% | 0.70 (0.46, 1.08) | 0.11 | – | – |
| **HIV outcome 1:** Proportion of individuals reporting a new HIV diagnosis (N=5683) | | | | | | |
| Door-to-door only | 39/1542 | 2.5% | | | | |
| Door-to-door plus household/community collection | 99/4141 | 2.4.% | 1.01 (0.49, 2.09) | 0.98 | 1.06 (0.50, 2.25) | 0.88 |
| **HIV outcome 2:** Undetectable viral load among PLHIV (N=830) | | | | | | |
| Door-to-door only | 138/211 | 65.4% | | | | |
| Door-to-door plus household/community collection | 371/619 | 59.9% | 0.79 (0.48, 1.30) | 0.36 | 0.77 (0.52, 1.14) | 0.20 |

PLHIV=people living with HIV; PrEP=pre-exposure prophylaxis; VMMC=voluntary medical male circumcision.

Adjusted odds ratios are not presented for the specific linkage outcomes (Linkage outcomes 1–3) due to the small number of cases. Adjusted models are adjusted for district and respondent age in 10-year groups, sex, and educational attainment (no or primary education, some secondary education, completed secondary education). All models adjusted for study community using a random effect.

(95% CI: 0.64-1.51), p=0.92). Independent of the distribution model and among all who received a self-test kit (1,831), use of the collected self-test kit did not differ by whether the kit was received door-to-door or elsewhere; 896/1,325 (67.6%) participants self-tested and received a kit door-to-door while 333/506 (65.8%) self-tested and received a kit by other means (aOR 1.08 (95% CI: 0.86-1.35), p=0.50). Similarly, at cluster-level self-testing uptake did not differ by whether the kit was received door-to-door; there was a -2% change in HIVST uptake (95% CI -10,+7, p=0.72) in head-man units implementing the combined HIVST distribution model compared with headman units conducting door-to-door distribution only. Linkage to post-test services was reported by 217/871 (24.9%) and 101/358 (28.2%) participants where combined or door-to-door distribution models were implemented,respectively (aOR 0.92 (95% CI: 0.66-1.27), p=0.60). Lastly, participants reporting an undetectable viral load was 371/619 (59.9%) and 138/211 (65.4%) among participants where combined or door-to-door distribution models were implemented, respectively (aOR 0.77 (95% CI: 0.52-1.14), p=0.20).

## Effect of levels of community involvement in planning distribution programmes on outcomes

Study outcomes did not differ by levels of community involvement (Table 7).

New HIV diagnosis did not differ across groups, with reports by 11/588 (1.9%) participants in the low group, 127/3,080 (4.1%) in the medium group, (aOR 1.98 (95% CI:0.67-5.85), p=0.22) and 73/2,015 (3.6%) in the high group (aOR 1.73 (0.65-4.59), p=0.27).

**Table 7. Comparison of outcomes by levels of community involvement in planning distribution programmes.**

| Outcome | | | Odds ratio (95% CI) | p-value | Adjusted odds ratio (95% CI) | p-value |
|---|---|---|---|---|---|---|
| | n/n | % | | | | |
| **Uptake outcome:** Self-reported self-testing uptake (N=5683) | | | | | | |
| Low involvement | 131/588 | 22.3% | | | | |
| Medium involvement | 650/3080 | 21.1% | 0.84 (0.41, 1.72) | 0.64 | 0.56 (0.29, 1.07) | 0.08 |
| High involvement | 448/2015 | 22.2% | 0.90 (0.43, 1.89) | 0.78 | 1.09 (0.56, 2.11) | 0.80 |
| **Combined linkage outcome:** Self-reported linkage to confirmatory testing, VMMC and PrEP (N=1229) | | | | | | |
| Low involvement | 34/131 | 26.0% | | | | |
| Medium involvement | 170/650 | 26.2% | 1.01 (0.49, 2.07) | 0.98 | 0.68 (0.33, 1.40) | 0.29 |
| High involvement | 114/448 | 25.4% | 0.80 (0.38, 1.71) | 0.57 | 0.60 (0.31, 1.17) | 0.14 |
| **Linkage outcome 1:** Self-reported linkage to confirmatory testing (N=1229) | | | | | | |
| Low involvement | 2/131 | 1.5% | | | | |
| Medium involvement | 22/650 | 3.4% | 2.26 (0.52, 9.73) | 0.27 | – | – |
| High involvement | 7/448 | 1.6% | 1.02 (0.21, 4.99) | 0.98 | – | – |
| **Linkage outcome 2:** Self-reported linkage to VMMC (N=1229) | | | | | | |
| Low involvement | 12/131 | 9.2% | | | | |
| Medium involvement | 59/650 | 9.1% | 1.12 (0.38, 3.25) | 0.84 | – | – |
| High involvement | 51/448 | 11.4% | 0.97 (0.32, 2.97) | 0.96 | – | – |
| **Linkage outcome 3:** Self-reported linkage to PrEP (N=1229) | | | | | | |
| Low involvement | 22/131 | 16.8% | | | | |
| Medium involvement | 113/650 | 17.4% | 1.03 (0.51, 2.09) | 0.93 | – | – |
| High involvement | 79/448 | 17.6% | 0.91 (0.43, 1.94) | 0.81 | – | – |
| **HIV outcome 1:** Proportion of individuals reporting a new HIV diagnosis (N=5683) | | | | | | |
| Low involvement | 8/588 | 1.4% | | | | |
| Medium involvement | 90/3080 | 2.9% | 2.08 (0.70, 6.17) | 0.19 | 1.94 (0.52, 7.21) | 0.32 |
| High involvement | 40/2015 | 2.0% | 1.11 (0.35, 3.56) | 0.86 | 1.25 (0.38, 4.13) | 0.71 |
| **HIV outcome 2:** Undetectable viral load among PLHIV (N=830) | | | | | | |
| Low involvement | 55/86 | 64.0% | | | | |
| Medium involvement | 308/494 | 62.3% | 0.86 (0.41, 1.79) | 0.68 | 0.76 (0.36, 1.62) | 0.48 |
| High involvement | 146/250 | 58.4% | 0.77 (0.35, 1.67) | 0.51 | 0.94 (0.49, 1.80) | 0.84 |

PLHIV = people living with HIV; PrEP = pre-exposure prophylaxis; VMMC = voluntary medical male circumcision.

Adjusted odds ratios are not presented for the specific linkage outcomes (Linkage outcomes 1–3) due to the small number of cases. Adjusted models are adjusted for district and respondent age in 10-year groups, sex, and educational attainment (no or primary education, some secondary education, completed secondary education). All models adjusted for study community using a random effect.

There were no differences in HIVST uptake across community involvement groups, with 131/588 (22.3%) participants in the low involvement group, 650/3,080 (21.1%) in the medium involvement group, (aOR 0.56 (0.29-1.07), p=0.08) and 448/2,015 (22.2%) in the high involvement group (aOR 1.09 (0.56-2.11), p=0.80), reporting HIVST uptake. There were no differences in linkage to post-test services across groups, with 131/588 (22.3%) participants linking in the low group, 650/3,080 (21.1%) in the medium group, (aOR 0.56 (0.29-1.07), p=0.08) and 448/2,015 (22.2%) in the high group (aOR 1.09 (0.56-2.11), p=0.80). Undetectable viral load which was 55/86 (64.0%) among participants in the low group, 308/494 (62.3%) in the medium group, (aOR 0.76 (0.36-1.62), p=0.48) and 146/250 (58.4%) in the high group (aOR 0.94 (0.49-1.80), p=0.84) did not differ. Finally, there were no statistically significant associations between (i) distribution model and levels of community involvement (p=1.0), (ii) distribution model and community cohesion (p=0.13), and (iii) levels of community involvement and community cohesion (p=0.15).

## Discussion

In this mixed-methods study, we examined the effects of community cohesion, HIVST distribution models and levels of community involvement in planning distribution programmes on: (i) self-reported new HIV diagnosis (ii) self-reported HIVST uptake; (iii) self-reported linkage to confirmatory testing, VMMC and PrEP; and (iv) viral load, among headman units conducting community-led HIVST distribution. We found the proportion of participants reporting a new HIV diagnosis increased with evidence of community cohesion and there was a dose response, with 1.8%, 2.1% and 3.2% in low-, medium-, and high-cohesion groups respectively. The type of HIVST distribution models implemented by headman units did not affect outcomes, nor did levels of community involvement.

The finding of self-reported new HIV diagnosis increasing with community cohesion is in line with our hypothesis that cohesive communities would achieve better outcomes. Similar evidence was found in the parent trial; in high cohesion communities the odds of new HIV diagnosis was greater in the community-led arm than in the comparison arm (OR 2.06 (95% CI: 1.03-4.19), p = 0.04) [21].

There is some evidence that cohesive communities are close-knit with a sense of social identity, belonging and under-standing of each other's health needs. In addition, working together provides space to confront myths, misconceptions, and stereotypes about people living with HIV (PLHIV) thereby reducing HIV stigma. In Zimbabwe, participation in community groups facilitated linkage to HIV prevention, care and treatment services and was associated with lower levels of HIV stigma [22] (the adverse effects of stigma on uptake of HIV-related services, health outcomes and quality of life among PLHIV has been documented [23–25]). In cohesive communities, community members' concern for good health extends beyond the individual to other members. Guided by group-based norms and values - the belief that "together we achieve better and more" (collective efficacy) - cohesive communities would collaborate effectively to achieve a common goal, eliminating new HIV infections through community-led HIVST distribution. In such headman units, U = U campaigns could have appealed to community members and motivated testing. As a result, HIVST distributors knew who to target with HIVST (active case finding), furthermore, good existing social relationships and trust for the distributor mediated community members' acceptance of the offer of kits [15], resulting in people who would not otherwise test, opting to test.

The lack of differences in self-testing uptake by community cohesion and distribution model could be attributed to each headman unit working together to design and refine ways of distributing kits in their setting. Such models would overcome context-specific barriers to achieve optimal uptake.

In this study, cohesion was associated with higher reports of new HIV diagnosis and the HIVST distribution period was associated with higher ART initiation rates at health facilities with or without HIVST in their catchment areas [18]. It is likely those newly diagnosed and initiated on ART under WHO's "Treat All" policy [26] may not have achieved undetectable viral load by the time of the survey (3–4 months after distribution), therefore no differences in undetectable viral load were found across cohesion groups. In their study Ali and colleagues found the median time to achieve viral load suppression after initiation of ART to be 181 days (CI: 140.5-221.4) [27]. This may explain why the other community measures (distribution models and levels of community involvement) had no effect on undetectable viral load.

In the parent trial, we found similar outcomes between the community-led and the paid distributor arms; linkage outcomes and reports of new HIV diagnosis in the intervention arm were comparable with those using a paid distributor model [18], showing communities were able to develop effective models. Our process evaluation data (not presented here), as well as other studies, however, suggest there may be some barriers to linkage to post-test services that still need to be addressed for self-testers such as: the belief that linkage to post-test services is unnecessary for HIV negative people [28–30], poor or inaccurate knowledge of PrEP [30–33] and VMMC [34,35], fear of pain during the VMMC procedure [34–36] and long distances to health facilities [37].

While incentives were not provided in this study, healthcare workers felt provider incentives would improve linkage [38]; though evidence on the impact of incentives remains variable [39]. These barriers may have affected linkage to post-test services in headman units for each of the community factors.

The strengths of this study included the use of robust methods for documenting and analysing how the community-led intervention was implemented. This study also adds to evidence on the positive effects of community cohesion on positive health behaviours and outcomes [15,17]. While most studies on social cohesion consider cohesion at the individual level [15,17], this study attempted to measure community cohesion systematically by observing levels of community involvement using a structured observation tool at the community-level. Our measure, levels of community involvement, relates to 4 out of 6 characteristics of Campbell and colleagues' [22] conceptualisation of HIV competent communities, namely: (i) critical thinking about obstacles to health-enhancing behaviour change, and discussions of locally realistic strategies for tackling these; (ii) promoting a sense of local ownership and responsibility for contributing to efforts that combat HIV/AIDS (iii) fostering a sense of solidarity and common purpose in confronting HIV/AIDS and (iv) identification of individual and group strengths for this challenge [22]. Furthermore, use of this measure was moderated by independent scoring by two researchers and resolving discrepancies through consensus.

Limitations of the study include the reliance on self-reported outcomes. While it is possible willingness to self-report varied by community cohesion, this is unlikely as this factor seemed to affect new HIV diagnosis alone but not other outcomes such as HIVST uptake or linkage. ACASI and laboratory testing of DBS samples to measure viral load were also used to minimise self-reporting bias. Levels of community involvement and community cohesion are related constructs, and the former may be a feature of community cohesion. However, the community involvement variable was weakly associated with the validated community cohesion measure - possibly due to the small sample size of 20 communities - and the associations between community cohesion and new HIV diagnosis were in line with our initial hypothesis. Although we systematically measured community involvement using participant observation, this was not a validated approach.

In summary, we found community-led HIVST distribution a feasible andacceptable way to distribute HIVST kits among rural Zimbabwean communities, accommodating flexibility in distribution models and varying levels of community involvement. They achieve outcomes similar to those in programmes implemented by professionally supported, paid distributors. Community cohesion in rural settings was associated with an increase in self-reported new HIV diagnoses. This suggests more cohesive communities may be better able to identify those most at risk of undiagnosed HIV infection, and people who need to test are likely to accept self-test kits from fellow community members, under a programme endorsed by their community leaders and in an environment where HIV can be freely discussed. Regardless of levels of community cohesion, future community-led HIVST programmes may learn from this approach by enhancing messaging on HIVST and post-test services; addressing related knowledge gaps; and confronting HIV-related myths, misconceptions, and stereotypes (stigma reduction interventions).

With time, communities implementing community-led interventions may become more cohesive. Further, implementation costs may gradually decrease as communities become more experienced [18], meaning community-led models may become more sustainable. Uniquely, community-led models foster programme ownership and empowerment. Communities can learn from and adopt community-led approaches to benefit other health priorities. Finally, policymakers could leverage this evidence by integrating and scaling-up community-led HIVST distribution models into existing health frameworks and national HIV/AIDS strategies. Alignment with this will facilitate decentralising services to the community level, effectively engaging populations who are currently underserved.

## Supporting information

**S1 data. S1_Data.dta.**
(DTA)

**S2 Data. S2_Data.pdf.**
(PDF)

## Acknowledgments

The authors appreciate the support of the Zimbabwe Ministry of Health and Child Care and the STAR Initiative Consortium partners, and the contribution of field researchers and participants, who all made this study possible.

## Author contributions

**Conceptualization:** Melissa Neuman, Miriam Taegtmeyer, Cheryl C Johnson, Karin Hatzold, Elizabeth L Corbett, Frances M Cowan, Euphemia L Sibanda.

**Data curation:** Mary K Tumushime, Nancy Ruhode, Melissa Neuman, Constancia Watadzaushe, Miriam Mutseta, Miriam Taegtmeyer, Euphemia L Sibanda.

**Formal analysis:** Mary K Tumushime, Nancy Ruhode, Melissa Neuman, Constancia Watadzaushe, Miriam Taegtmeyer, Euphemia L Sibanda.

**Funding acquisition:** Miriam Taegtmeyer, Cheryl C Johnson, Karin Hatzold, Elizabeth L Corbett, Frances M Cowan, Euphemia L Sibanda.

**Investigation:** Mary K Tumushime, Nancy Ruhode, Constancia Watadzaushe, Miriam Mutseta.

**Methodology:** Mary K Tumushime, Nancy Ruhode, Constancia Watadzaushe, Miriam Mutseta, Miriam Taegtmeyer, Elizabeth L Corbett, Frances M Cowan, Euphemia L Sibanda.

**Project administration:** Mary K Tumushime, Nancy Ruhode, Constancia Watadzaushe, Miriam Mutseta, Euphemia L Sibanda.

**Writing – original draft:** Mary K Tumushime, Nancy Ruhode, Melissa Neuman.

**Writing – review & editing:** Mary K Tumushime, Nancy Ruhode, Melissa Neuman, Miriam Mutseta, Miriam Taegtmeyer, Cheryl C Johnson, Frances M Cowan, Euphemia L Sibanda.

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
