## [Decision Letter · Decision Letter 0]

9 Aug 2024

PGPH-D-24-00701

What role do community-level factors play in HIV self-testing uptake, linkage to services and HIV-related outcomes? A mixed methods study of community-led HIV self-testing programmes in rural Zimbabwe

Dear Dr. Tumushime,

Thank you for submitting your manuscript to PLOS Global Public Health. After careful consideration, we feel that it has merit but does not fully meet PLOS Global Public Health’s publication criteria as it currently stands. Therefore, we invite you to submit a revised version of the manuscript that addresses the points raised during the review process.

Please note that we have only been able to secure a single reviewer to assess your manuscript. We are issuing a decision on your manuscript at this point to prevent further delays in the evaluation of your manuscript. Please be aware that the editor who handles your revised manuscript might find it necessary to invite additional reviewers to assess this work once the revised manuscript is submitted. However, we will aim to proceed on the basis of this single review if possible. 

Please carefully address the reviewers comments and revise your manuscript accordingly. 

We look forward to receiving your revised manuscript.

Kind regards,

Jennifer Tucker, PhD

Staff Editor

Journal Requirements:

1. We ask that a manuscript source file is provided at Revision. Please upload your manuscript file as a .doc, .docx, .rtf or .tex.

2. In the online submission form, you indicated that Data are available upon request. De-identified data are available from the corresponding author upon request. 

a. In a public repository, 

b. Within the manuscript itself, or 

c. Uploaded as supplementary information.

Additional Editor Comments (if provided):

Reviewers' comments:

Reviewer's Responses to Questions

**Comments to the Author**

1. Does this manuscript meet PLOS Global Public Health’s publication criteria ? Is the manuscript technically sound, and do the data support the conclusions? The manuscript must describe methodologically and ethically rigorous research with conclusions that are appropriately drawn based on the data presented.

Reviewer #1: Yes

2. Has the statistical analysis been performed appropriately and rigorously?

Reviewer #1: Yes

3. Have the authors made all data underlying the findings in their manuscript fully available (please refer to the Data Availability Statement at the start of the manuscript PDF file)?

Reviewer #1: Yes

4. Is the manuscript presented in an intelligible fashion and written in standard English?

Reviewer #1: Yes

5. Review Comments to the Author

Reviewer #1: The manuscript provides findings that are very relevant to the current programmatic gaps and provides insights to improve community testing specially for men who are still lagging behind in the first 95.

I however suggest that the author clearly defines who the community is in the context of this paper, and the characteristics of the community described for purposes of evaluating the value added to the body of knowledge on the subject matter.

The author should also define what cohesion means in the context of the manuscript since its one of the variables used in the analysis.

It would be good to have the author explain if these findings are still relevant to the current Zimbabwe context noting that the self testing strategy is now scaled up beyond the setting described in this paper.

It will be important get clarification from the author about the gap that self testing will address in the current community described in the manuscript, is it a gap filler, is it a strategy for the hard to reach or it is another prevention option for the populations in need.

The author should explain how this self testing strategy will become part of the policy for future scale up and how sustainability can be achieved through embedment in policy

6. PLOS authors have the option to publish the peer review history of their article (what does this mean? ). If published, this will include your full peer review and any attached files.

**Do you want your identity to be public for this peer review?** For information about this choice, including consent withdrawal, please see our Privacy Policy .

Reviewer #1: **Yes: ** Helgar Musyoki, technical Advisor Key Populations ,The Global fund ,Geneva

---

## [Decision Letter · Decision Letter 1]

20 Feb 2025

Do community-level factors play a role in HIV self-testing uptake, linkage to services and HIV-related outcomes? A mixed methods study of community-led HIV self-testing in rural Zimbabwe

PGPH-D-24-00701R1

Dear Ms Tumushime,

We are pleased to inform you that your manuscript 'Do community-level factors play a role in HIV self-testing uptake, linkage to services and HIV-related outcomes? A mixed methods study of community-led HIV self-testing in rural Zimbabwe' has been provisionally accepted for publication in PLOS Global Public Health.

Best regards,

Jeffrey William Eaton

Academic Editor

Reviewer Comments (if any, and for reference):

Reviewer's Responses to Questions

**Comments to the Author**

1. If the authors have adequately addressed your comments raised in a previous round of review and you feel that this manuscript is now acceptable for publication, you may indicate that here to bypass the “Comments to the Author” section, enter your conflict of interest statement in the “Confidential to Editor” section, and submit your "Accept" recommendation.

Reviewer #1: All comments have been addressed

2. Does this manuscript meet PLOS Global Public Health’s publication criteria ? Is the manuscript technically sound, and do the data support the conclusions? The manuscript must describe methodologically and ethically rigorous research with conclusions that are appropriately drawn based on the data presented.

Reviewer #1: Yes

3. Has the statistical analysis been performed appropriately and rigorously?

Reviewer #1: (No Response)

4. Have the authors made all data underlying the findings in their manuscript fully available (please refer to the Data Availability Statement at the start of the manuscript PDF file)?

Reviewer #1: Yes

5. Is the manuscript presented in an intelligible fashion and written in standard English?

Reviewer #1: Yes

6. Review Comments to the Author

Reviewer #1: All comments have been addressed

7. PLOS authors have the option to publish the peer review history of their article (what does this mean? ). If published, this will include your full peer review and any attached files.

**Do you want your identity to be public for this peer review?** For information about this choice, including consent withdrawal, please see our Privacy Policy .

Reviewer #1: **Yes: ** Helgar Musyoki
